# Immune-Related Gene Profiles and Differential Expression in the Grey Garden Slug *Deroceras reticulatum* Infected with the Parasitic Nematode *Phasmarhabditis hermaphrodita*

**DOI:** 10.3390/insects15050311

**Published:** 2024-04-26

**Authors:** Muhammad Hafeez, Rory Mc Donnell, Andrew Colton, Dana Howe, Dee Denver, Ruth C. Martin, Man-Yeon Choi

**Affiliations:** 1USDA-ARS, Horticultural Crops Disease and Pest Management Research Unit, Corvallis, OR 97330, USA; muhammad.hafeez@oregonstate.edu; 2Department of Horticulture, Oregon State University, Corvallis, OR 97331, USA; ruth.martin@oregonstate.edu; 3Department of Crop and Soil Science, Oregon State University, Corvallis, OR 97331, USA; rory.mcdonnell@oregonstate.edu (R.M.D.); andrew.colton@outlook.com (A.C.); 4Department of Integrative Biology, Oregon State University, Corvallis, OR 97331, USA; howeda@oregonstate.edu (D.H.); dee.denver@oregonstate.edu (D.D.)

**Keywords:** grey garden slug, *Deroceras reticulatum*, immunity, *Phasmarhabditis hermaphrodita*, nematode infection, RNA-Seq

## Abstract

**Simple Summary:**

Slugs and snails are widely distributed in natural humid habitats, and they are a worldwide problem in agriculture. The grey garden slug, a common terrestrial species, is considered the most severe pest of vegetables and field crops. Currently, the most common slug control methods rely on chemical pesticides, which can be damaging to the environment and human health. Nematodes are important natural enemies of slugs, and one species has been used as a commercial product for three decades. This study investigated and analyzed the differential gene expression profiles between nematode-infected slugs and uninfected slugs and identified the genes associated with immunity in the grey garden slug. The results provide a starting point for understanding the molecular mechanism of immune genes and physiological pathways, facilitating the identification of biological targets for slug management strategies in the field.

**Abstract:**

The grey garden slug (*Deroceras reticulatum*), a common terrestrial slug native to Europe with a global distribution including North America, is commonly considered the most severe slug pest in agriculture. The nematode *Phasmarhabditis hermaphrodita*, which has been used in the U.K. and Europe as a commercial biocontrol agent since 1994, has also recently been collected in Oregon and California and has long been considered a candidate biocontrol agent for slug management in the U.S. In this study, we report differential gene expressions in nematode-infected slugs using RNA-seq to identify slug immune-related genes against nematodes. Comparison of gene expression levels between the whole bodies of a nematode-infected slug (N-S) and an uninfected control slug (C-S) revealed that there were a total of 39,380 regulated unigenes, of which 3084 (3%) were upregulated and 6761 (6%) were downregulated at greater than 2-fold change (FC > 2) in the nematode-infected slug. To further investigate the biological functions of differentially expressed genes (DEGs), gene ontology (GO) and functional enrichment analysis were performed to map the DEGs to terms in the GO, eukaryotic ortholog groups of proteins (KOG) and Kyoto Encyclopedia of Genes and Genome Pathway (KEGG) databases. Among these DEGs, approximately 228 genes associated with immunity or immune-related pathways were upregulated 2-fold or more in the N-S compared to C-S. These genes include toll, Imd, JNK, scavenger receptors (SCRs), C-type lectins (CTLs), immunoglobulin-like domains, and JAK/STAT63 signaling pathways. From the RNA-seq results, we selected 18 genes and confirmed their expression levels by qRT-PCR. Our findings provide insights into the immune response of slugs during nematode infection. These studies provide fundamental information that will be valuable for the development of new methods of pest slug control using pathogenic nematodes in the field.

## 1. Introduction

Slugs and snails (Mollusca: Gastropoda) are widely distributed in natural humid habitats and pose a significant threat to various agricultural and horticultural crops worldwide [1,2]. The grey garden slug, *Deroceras reticulatum* (Stylommatophora: Agriolimacidae), a common terrestrial slug native to Europe with a global distribution including North America, is commonly considered the most severe pest of vegetables and field crops [3,4]. *D. reticulatum* poses serious problems at various stages of field vegetable production. In severe cases, entire fields may need to be replanted, resulting in significant economic losses [5]. Furthermore, contamination from slug mucus and feces can adversely impact the quality of the harvested crop [6]. The grey garden slug is primarily nocturnal, with its highest level of activity occurring shortly after dusk [7]. However, when the weather is calm, damp, and overcast, it may also become active during the daytime.

Currently the most common slug control methods rely on chemical pesticides that are primarily mixed into pellet bait-based products [8,9]. For crop protection purposes, the efficacy of these products is highly variable. There are also environmental risks associated with chemical residue in soil and water and their effect on non-target arthropods and human health [10,11,12]. These issues along with growing public opposition to pesticides and a greater appreciation of sustainable control methods necessitate the development of biologically based environmentally friendly options for slug control. One such approach is the use of parasitic nematodes as biological control agents. The biological interaction between a parasite and its host is complicated by the physiological complexity and the two immune systems.

Numerous genes are linked to the immune response in invertebrates, both at the cellular and humoral levels [13,14,15]. A recent approach involving genome-wide analysis has been employed to identify genes associated with immunity and to investigate the molecular mechanisms underlying interactions between hosts and microorganisms in insects [14,15,16,17]. Specifically, the utilization of whole genome mRNA sequencing (RNA-seq or transcriptome sequencing) has provided a comprehensive understanding of the repertoire of immune-related genes in non-model insects [18,19]. This technology has emerged as a potent tool for analyzing differential gene expression [20,21,22,23]. In gastropods, it is reasonable to assume that different genes are associated with the slug’s immune system to protect against pathogen and parasite attack, including parasitic nematodes.

There are more than 100 nematode species associated with slugs and snails [24] serving as definitive, intermediate, or necromenic hosts [24,25]. Among these, 47 nematode species from eight families utilize mollusks as definitive hosts [4], some of which are lethal parasites, and thus offer potential as biological control agents due to their natural associations with terrestrial gastropods [26,27]. Notably, only the facultative parasitic nematodes *Phasmarhabditis hermaphrodita* and *Phasmarhabditis californica* (Rhabditida: Rhabditidae) have been developed as commercial biological agents of pest slugs and snails in the U.K. and Europe [9]. However, both of these species and other species of *Phasmarhabditis* have also been found throughout the west coast of the United States [28,29,30,31,32], and *P. californica* has been found in Alberta, Canada [29]. Previous studies reported that *P. hermaphrodita* nematodes travel to the shell cavity through the back of the mantle after finding a slug host [4,33]. Subsequently, the larvae undergo development into self-fertilizing hermaphrodites commencing the reproductive phase [4,33]. Symptoms of infection, including a swelling of the mantle and shell ejection, start to show within 4–21 days of the initial infection [4], allowing the nematodes to feed and reproduce. Dauer larvae crawl into the soil to find a new host when their food supply becomes exhausted. This study aims to identify potential biological targets, particularly immune-related genes, of *D. reticulatum*, after infection by the nematode, *P. hermaphrodita*, since most physiological and genotypic actions of the slug will be related to its immune response during the parasitic process.

In this study, as a first step, we investigated and analyzed the gene expression profiles of the nematode-infected slugs using Illumina RNA-Seq, compared them to the uninfected slugs, and explored the differentially expressed genes involved in immunity and related pathways in the grey garden slug. We conducted a de novo transcriptome analysis to identify differentially expressed immune-related genes/pathways from the nematode-infected slug (N-S) and uninfected control slug (C-S). These annotations provide a starting point for investigating the molecular mechanisms of immune-related genes in *D. reticulatum* and provide a valuable resource for further research into the immune specific functions and pathways of *D. reticulatum*. Our current research to understand the molecular mechanisms of immune genes and physiological pathways facilitates the identification of biological targets for slug management strategies in the field.

## 2. Material and Methods

### 2.1. Slugs and Infection of the Nematodes

*Deroceras reticulatum* were collected from various grass seed production fields throughout the Willamette Valley, OR, USA. Field-collected slugs were identified and confirmed as *D. reticulatum* using [34,35]. Slugs were not cultured in the laboratory. We used field-collected slugs and nematode-infected slugs according to previously published methods and guidelines [27,32,36]. On return to the laboratory, slugs were placed into plastic containers (35.9 cm × 20 cm × 12.4 cm) that were lined with a single sheet of wet paper towel (Bounty Select-a-Size). Slices of organic carrot were provided as food, and the paper towel and carrot were replaced three times weekly. Thirty slugs were placed in each container, and containers were stored in a growth chamber (Thermo Scientific Precision Model 818) under 18 °C and L/D 12:12.

Infection trial protocol largely followed [27], except only a single rate (40,000 mixed stage nematodes) of *P. hermaphrodita* was utilized. Eight adult slugs were used for each of five nematode and water control replicates. After 7 days, three slugs from each replicate were removed for the current study. The mortality of the remaining slugs in the infection trial was recorded daily for 14 days. After this time, there was 100% slug mortality in all nematode replicates and 0% mortality in the water control replicates. The slugs samples were then routed to various next steps as below (Figure 1).

### 2.2. Total RNA Preparation

Total RNA was isolated from the whole body of a single adult slug infected with nematodes and a noninfected control slug using the Purelink Total RNA Purification System (Thermo Fisher Scientific, Waltham, MA, USA). RNA was treated with TURBO^TM^ DNase (Thermo Fisher Scientific) for 30 min at 37 °C to eliminate genomic DNA, according to the manufacturer’s instructions (Figure 1). RNA was further purified by using the RNeasy MinElute Cleanup Kit (Qiagen, Germantown, MD, USA) and eluted in 20 μL of RNA storage solution. The quantity of the RNA was assessed using a NanoDrop Spectrophotometer ND-2000 (Thermo Fisher Scientific). The six RNA samples (N-S and C-S per 3 replicates) were then sent to Psomagen (Rockville, MD, USA) for RNA quality analysis using an Agilent 2100 Bioanalyzer (Agilent Technologies, Santa Clara, CA, USA). To maximize RNA quality, only samples with an RNA integrity number (RIN) value of 7 or greater were used for the next step.

### 2.3. cDNA Library Preparation and RNA Illumina Sequencing

The cDNA libraries were prepared using a TruSeq Stranded Total RNA Library Prep Kit (Illumina Inc., San Diego, CA, USA), and the Illumina sequencing was performed by Psomagen using the Illumina NovaSeq6000 platform. Briefly, ribosomal RNA was removed from total RNA, and the remaining RNA was purified, fragmented, and primed for cDNA synthesis. The first-strand cDNA was synthesized by priming the RNA fragments with random hexamers and by transcribing with reverse transcriptase. RNA template of the first strand cDNA was replaced by incorporating dUTP in place of dTTP to generate second strand cDNA. The cDNA then underwent an end repair process, adenylation of 3′ ends, and subsequent ligation of the adapter. The adaptor-ligated library was purified and enriched with PCR to create the final cDNA library. The quantified and qualified libraries were sequenced using an Illumina NovaSeq6000. For cluster generation, the library was loaded into a flow cell where fragments were captured on a lawn of surface-bound oligos complementary to the library adapters. Each fragment was then amplified into distinct clonal clusters through bridge amplification. When cluster generation was complete, the templates were ready for sequencing.

### 2.4. De Novo Transcriptome Assembly

The raw sequence reads generated by Illumina sequencing were checked by FastQC for quality control using an integrated primary analysis software program called RTA (Real Time Analytics, version 1). We manually checked all putative hits, looking for the presence of the candidate genes and characteristic domains. The candidate transcripts were further searched against the NCBI database ‘https://blast.ncbi.nlm.nih.gov/Blast.cgi (accessed on 16 April 2024)’. Trimmomatic (version 0.32) [37] was used to remove adaptors and low-quality reads. Overlapping high-quality reads were de novo assembled to create longer contiguous fragments (contigs) using Trinity (version r2014-07-17) [38]. Transcript abundance was estimated using RSEM (version 1.2.15) to generate FPKM (fragments per kilobase per million reads).

### 2.5. Functional Annotation

The functional annotation of the transcripts was determined by sequence similarity searches against NCBI non-redundant protein sequences (nr) database using BLASTx algorithm with a cut-off E-value of 1 × 10^−5^ run by Geneious 8.1.5 software (Biomatters Ltd., Auckland, New Zealand). Gene ontology (GO) terms were mapped to transcripts with BLAST hits to assign functional categories [39]. The BLAST hits were grouped into major gene families based on their putative functions in slug biology. The organism information obtained by BLAST hits was collected to confirm the transcript’s sequence similarity to closely related species.

### 2.6. Identification of Immunity-Related Genes

The search for immunity-related genes was based on sequence similarities to known sequences among closely related species in NCBI using BLASTx. cDNAs encoding full- or partial-length immune-related genes from the *D. reticulatum* transcriptome were found and sorted into different immune gene categories and related pathways from 11 different families and groups. Sequence features mostly focus on their similarity and integrity compared to known sequences.

### 2.7. Validation of DEGs Results by qRT-PCR

Based on the expression values as FPKM, DEGs were identified between the two samples (N-S and C-S) after transformation, normalization, and fold change (fc) comparisons. Quantitative real-time PCR (qRT-PCR) was conducted using the SYBR Green method in a StepOnePlus Real-Time PCR System (Applied Biosystems, Walthman, MA, USA). Total RNA was extracted from slug–control and slug–nematode treatments using the Purelink Total RNA Purification System (Thermo Fisher Scientific). cDNA templates were synthesized from 1 μg of total RNA using the Invitrogen SuperScript III First-Strand Synthesis SuperMix according to the manufacturer’s instructions. The qRT-PCR reaction mixtures were prepared in 20 μL with 1× PowerUp SYBR Green Master Mix (Thermo Fisher Scientific), 0.25 μM primer pairs, and cDNA template. The qRT-PCR reaction conditions were performed at 95 °C for 10 min; 40 cycles of 95 °C for 15 s and 60 °C for 1 min; followed by a melting curve analysis over the range of 60–95 °C with 0.3 °C/min increments, with specific primers listed in Appendix A. The *Rpt6* gene was selected as a reference gene (Appendix A). Three biological samples for each group were replicated.

### 2.8. Statistical Analysis

Student’s *t*-test was used to compare the relative mRNA expression levels between control and treatment groups using GraphPad Prism 7.0 (San Diego, CA, USA). A *p*-value of 0.05 or less between groups was considered significantly different.

## 3. Results

### 3.1. Illumina Sequencing and De Novo Assembly

NovaSeq6000 technology was used to sequence six cDNA libraries from the whole body of *D. reticulatum* to investigate the effect of pathogenic nematodes on the immune response of the grey garden slug (Figure 1). The total raw reads for the replicate control and nematode infected slugs are listed in Table 1. The total number of reads with over 92% validity after mapping are summarized in Table 1. Results indicated that all libraries were high-quality, with an overall percentage of Q20 (95.98%) and Q30 (93.75%) with an average 40% GC ratio (Table 1).

### 3.2. Functional Annotation of the Control Slug and Nematode-Infected Slug Unigenes

Six different public databases including GO, KEGG, Pfam, Swiss-Prot, eggnog, and NR were used to determine the functional annotation of unigene sequences shown in Table 2. The results from aligned unigene sequences revealed that 7786 (12.04%); 7767 (12.01%); 7787 (12.04%); 7049 (10.90%); 9618 (14.87%) and 13,285 (20.54%) unigenes were matched to the GO, KEGG, Pfam, Swiss-Prot, eggNOG, and NR protein databases, respectively (Table 2).

### 3.3. Differentially Expressed Genes in the Nematode-Infected Slug

To explore the changes in the gene expression of *D. reticulatum* infected with pathogenic nematodes (nematode-infected slug), a pairwise comparison was performed between the nematode-infected and the control slug libraries to determine the differentially expressed genes (DEGs). The screening threshold for DEGs in the infected slugs compared to the control slugs was set as genes having a fold-change greater than 1 and false discovery rate (FDR) value less than 0.001. Compared to the slug–control treatment, of the total 39,380 regulated unigenes, 29,307 genes (74.4%) had FC < 2. Among FC > 2 genes, 3084 (7.8%) were upregulated and 6761 (17.2%) were downregulated. We found 228 genes that were annotated to immunity or related pathways in the slug–nematode (Figure 2A). Volcano plots of DEGs were similar to the distribution of gene expression (FC > 2) of all DEGs in the nematode-infected slug compared (Figure 2B).

### 3.4. GO and KEGG Functional Classification of Nematode-Infected Slug Unigenes

Gene ontology was used to categorize a total of 14,518 unigenes into 51 subcategories under three main categories ‘biological process’, ‘cellular component’, and ‘molecular function’ (Figure 3). The ‘cellular component’ category was the most dominant with 8017 unigenes (55.2%) followed by ‘molecular function’ category with 3492 unigenes (24.1%) and the ‘biological process’ category with 3009 unigenes (20.7%) annotated in GO database.

Furthermore, we classified the unigenes in the Kyoto Encyclopedia of Genes and Genomes (KEGG) database and assigned them to five biological pathways: 2152 unigenes (39.0%) in metabolism, followed by 1518 unigenes (27.5%) in genetic information processing, 901 unigenes (16.3%) in cellular processes, 653 unigenes (11.8%) in environmental information processing and 331 unigenes (0.06%) in organismal systems (Figure 4). Within the total of 2152 metabolism genes, there were 409 unigenes (19.0%) for carbohydrates, 360 unigenes (16.7%) for amino acids, and 332 unigenes (15.4%) for lipids. Similarly, the second largest group, genetic information processing (1518 genes), were involved in translation with 624 (29%) followed by folding, sorting, and degradation 490 (22.8%), and transcription 311 (14.5%). Of genes involved in pathways of the organismal systems, most were in the immune system (84 unigenes) and the endocrine system (75 unigenes) (Figure 4).

### 3.5. Identification and Expression Pattern of Immunity-Related Genes in Response to Pathogenic Nematode Infection

A keyword search was employed for comprehensive analysis to identify immunity-related candidate genes expressed in response to pathogenic nematode infection in the grey garden slug *Deroceras reticulatum* from the BLASTx by searching the genome and by combining BLASTx search using NCBI ‘https://blast.ncbi.nlm.nih.gov/Blast.cgi?PROGRAM=blastx&PAGE (accessed on 16 April 2024)’. To increase the reliability of results, genes annotated as hypothetical or unknown proteins and genes with log2 fold change < 1 were filtered out. In total, 228 immunity-related genes (Log2 fold > 2) were identified and categorized into 10 different groups, such as signal transduction (86 genes), scavenger receptor (34 genes), immunoglobulin family protein (23 genes), sialic acid protein (22 genes), c-Jun N-terminal kinase (JNK) (21 genes), C-type lectin-like domain (CTLD) protein (19 genes), dual oxidase (Duox) (10 genes), toll-like receptor (TLR) (8 genes), bactericidal permeability-increasing protein (BPI) (3 genes), and CD109 antigen-like (2 genes), respectively. Among the immunity-related genes, signal transduction (86) were the most abundant DEGs followed by scavenger receptors (34) in the nematode-infected slug compared to the control slug (Figure 5 and Appendix A).

### 3.6. Validation of Differentially Expressed Immunity-Related Genes

From the RNA-seq results, we selected 13 immune-related genes that showed high levels of differential expression (FC > 2) in the nematode-infected slug (N-S) compared to the control slug (C-S) (Figure 6A). To validate the relative mRNA expression, qRT-PCR was performed to determine the relative gene expression ratios in the nematode-infected slugs and control slugs (Figure 6B). All selected genes were detected, and the relative expression ratios measured by qRT-PCR were slightly different from the RNA-seq results (Figure 6B). For example, the relative mRNA expression ratio of the C-type lectin gene was the highest from the RNA seq, but the sialic acid-binding lectin 3 gene had the highest expression ratio based on the qRT-PCR results, but the others were similarly expressed based on the qRT-PCR results.

## 4. Discussion

It is common for slugs to interact with their parasitic nematodes, both environmentally and evolutionarily. Nematoda, a phylum consisting of numerous species, has over 50% of its described members categorized as parasitic, positioning them as one of the most successful parasitic groups. In order to effectively invade hosts, parasitic nematodes need to possess the ability to shield themselves from the host’s immune system, ensuring their survival [40,41]. Nematodes utilize various tactics to interfere with the initiation and regular processes of host physiological reactions, many of which are involved in controlling innate immune responses [40]. Previously, extensive research has been carried out to investigate the correlation between nematodes and the immune system of the host [41,42]. However, no study has explored the impact of nematodes on the genes responsible for immunity and the associated pathways in slugs. The study of *D. reticulatum* derived innate immune reactions against nematode parasites is fundamental to a better understanding of the molecular, biochemical, and signaling pathways involved in their interactions.

Host–parasite interactions are highly complex. The host identifies the parasite as foreign and triggers immune responses and signaling in its own defense [13]. Typically, when animals are infected by parasites or pathogens, their immune systems become activated. Initially, pattern recognition proteins (PRPs) play a crucial role in identifying and marking these invaders [43,44]. Immune genes and related pathways, such as toll receptors, C-type lectins (CTLs), various oxidases, and MyD88-dependent pathways have been reported in many invertebrate pests including slugs [17,19,45,46,47,48,49,50]. These studies showed that the signal transduction pathways were immediately activated to produce antimicrobial chemicals. Four signaling pathways, namely the toll, Imd, JNK, and JAK/STAT63 [51] pathways, are considered important immune-related pathways in insects, as described previously [52,53]. In the current transcriptome analysis, signal transduction, toll pathways, and JNK pathways were regulated in the nematode-infected slug compared to the control slug group. Interestingly, signal transduction was the top regulated pathway (Figure 5 and Appendix A). Similar to our results, others have reported that the signal transduction, the toll and JNK pathways were modulated after exposure to a mycotoxin *destruxin A* and a nematode *Steinernema carpocapsae* in *Drosophila melanogaster, Bombyx mori*, and *Spodoptera frugiperda* [22,54,55]. Previously, it has been reported that scavenger receptors (SCRs), ‘glycoproteins’, C-type lectins (CTLs), a carbohydrate binding protein group, and immunoglobulin-like domains play a key role in cellular immune responses to protect invertebrates in response to pathogen infection [56,57,58].

In the present study, genes related to scavenger receptors, C-type lectins (CTLs), and immunoglobulin-like domains were upregulated in the nematode-infected slug compared to the water control group. Our results are in accordance with previous reports in which genes related to scavenger receptors, C-type lectins (CTLs), and immunoglobulin-like domains were regulated in different insect pests in response to pathogens [59,60,61]. Understanding immune-related pathways in response to nematode infection can facilitate the development of genetic manipulation to develop long-term management methods to control *D. reticulatum*. The 18 immune-related unigenes that were selected based on their role in various immune pathways and immune-related genes mainly belonged to the toll, Imd, JNK, scavenger receptor (SCR), C-type lectin (CTL), immunoglobulin-like domain, and JAK/STAT63 signaling pathways. These unigenes were validated using qRT-PCR, and the expression patterns of 12 of these immune-related unigenes were consistent with the transcriptome analysis.

We also found a total of 3084 upregulated and 6761 downregulated DE genes between the nematode-infected slug compared to the control group (Figure 2A). Among them, 228 immune DE genes (Log2 fold > 2) and related pathways were identified, and 11 immune pathways and related genes were categorized into different groups (Figure 5). The differentially expressed genes (DEGs) related to immune pathways were identified in various species of insects after exposure to nematodes and pathogenic fungi [22,55,62,63]. Thus, we believe that the regulation of immune pathways and related genes observed in this study indicates the promotion of the defense system of *D. reticulatum* by infecting parasitic nematodes. Immune systems must constantly evolve to remain effective in the face of both changes in the suite of pathogens to which they are exposed and the evolution of virulence mechanisms. These dynamics can result in a strong signature of adaptive evolution in genes involved in the immune response [64,65]. The innate immune system, which consists of cellular and humoral responses, is the first line of defense against pathogenic infections in invertebrates [66,67].

In conclusion, transcriptome profiling via RNA-Seq is a good approach for the assessment of transcript levels related to infection and immunity in *D. reticulatum*. RNA-Seq analysis of *D. reticulatum* infected by nematodes reveals transcriptional regulation of a large number of genes, many of which have not been shown previously to participate in immune processes against pathogenic infections in *D. reticulatum*. Many of these genes that are differentially regulated upon nematode infection are predicted to be involved in metabolic functions, immune functions, and stress response activities. In addition, we have identified *D. reticulatum* genes related to immune pathways with a potential anti-nematode role. Many of these pathways provide an excellent platform of candidate factors for the functional characterization of the *D. reticulatum* immune response against nematode complexes. Our findings not only offer deep insight into immunogenetics of *D. reticulatum* in response to nematodes, but also enhance current knowledge of interactions between host and pathogens. Future studies using the *D. reticulatum* immune system promise to reveal not only how pathogens evolve virulence but also how pathogens (nematodes) can synergize to exploit a common host.

## Figures and Tables

**Figure 1 insects-15-00311-f001:**
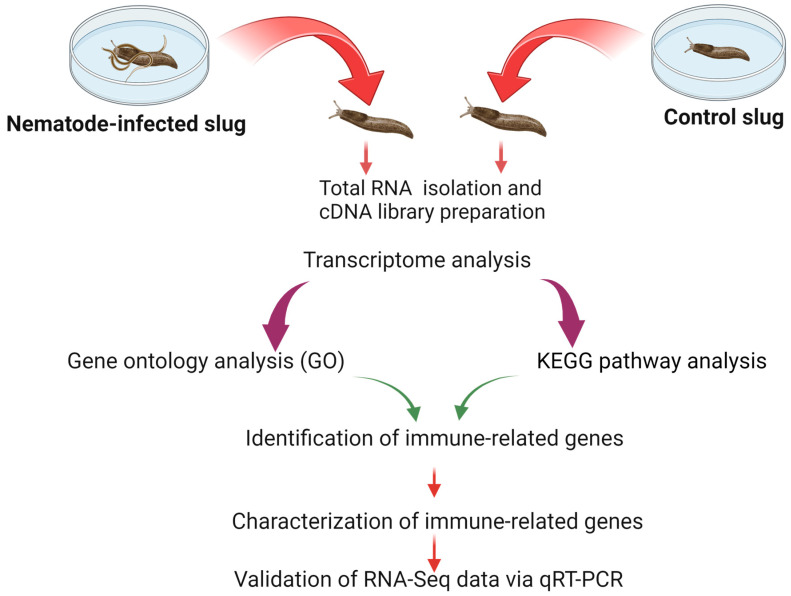
Outline of the experimental procedure of RNA transcriptome analysis and identification of genes associated with immunity and its pathways from nematode-infected and uninfected slugs.

**Figure 2 insects-15-00311-f002:**
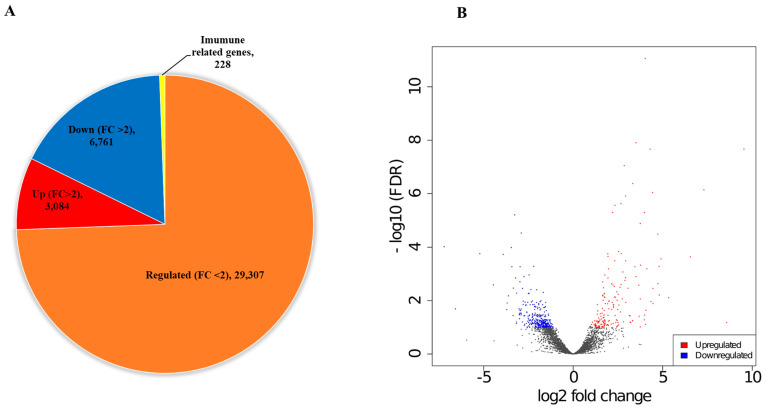
De novo transcriptome analysis of nematode-infected *Deroceras reticulatum*. A pie diagram of differentially expressed genes in nematode infected slug as compared to the uninfected slug. In the total number of regulated unigenes (39,380), genes with low log2 fold changes (FC < 2) are 29,307 (orange), 3084 genes are upregulated (red) and 6761 genes are downregulated (blue), with FC > 2 (**A**). Volcano plots of DEGs showing the distribution of gene expression plotted by FC > 2 for each gene in the nematode-infected slug compared to the control slug group. Red dots indicate upregulated genes, and blue dots indicate downregulated genes (**B**).

**Figure 3 insects-15-00311-f003:**
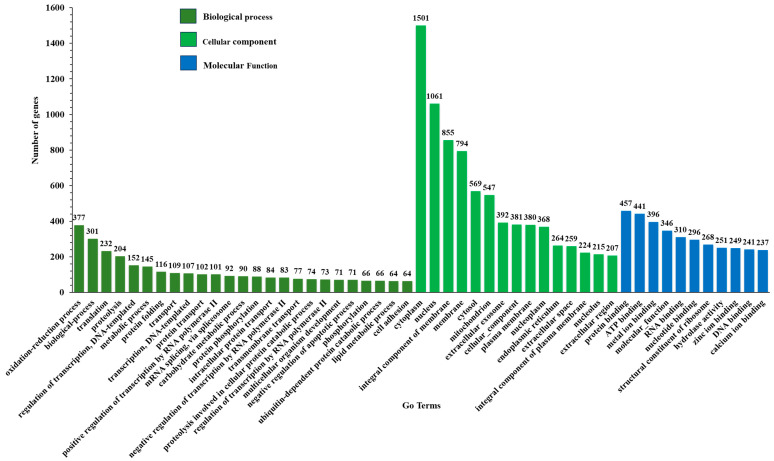
Gene ontology (GO)-enriched terms of differentially expressed genes (DEGs) of *Deroceras reticulatum* after infection with nematode. The *x*-axis lists the sub-GO terms under categories of biological process, cellular component, and molecular function. The *y*-axis is the number of DEGs involved in each term.

**Figure 4 insects-15-00311-f004:**
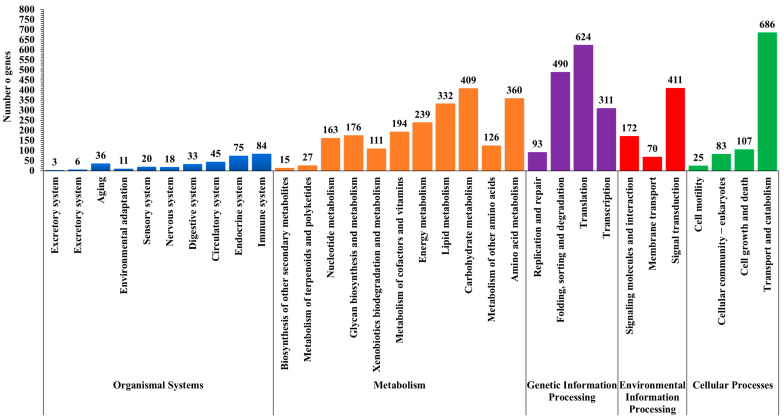
Kyoto Encyclopedia of Genes and Genomes (KEGG) pathway classification with 331 genes associated with organismal systems, 2152 with metabolism, 1518 with genetic information processing, 653 with environmental information processing, and 901 with cellular processes in *Deroceras reticulatum*.

**Figure 5 insects-15-00311-f005:**
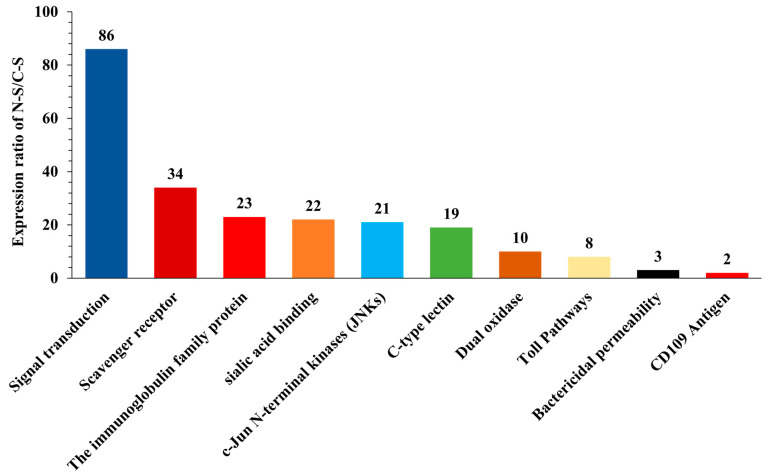
The total number of genes associated with immunity and immune-related pathways in the nematode-infected slug (N-S) compared to the uninfected control slug (C-S) group. To increase the reliability of the results, genes annotated as hypothetical or unknown and genes with log2 fold change < 1 were filtered out.

**Figure 6 insects-15-00311-f006:**
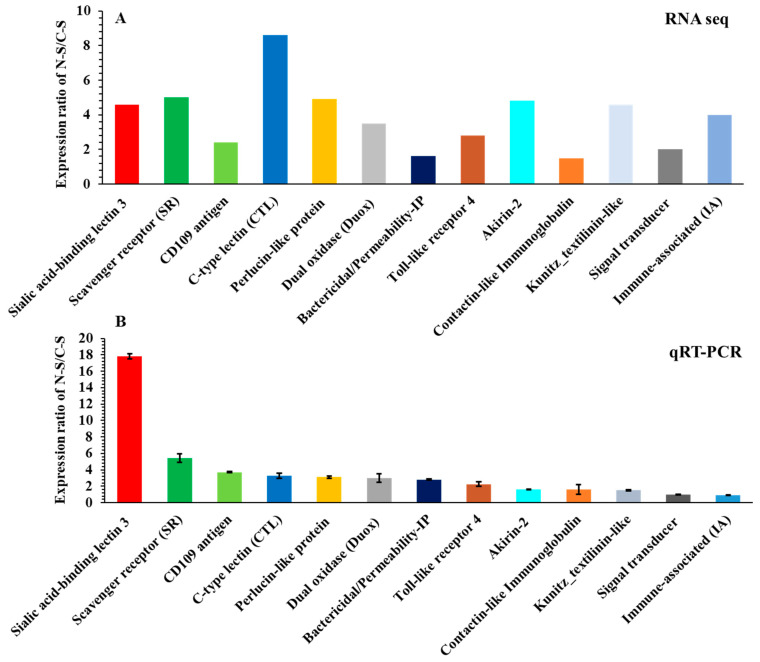
Expression of immunity-related genes in *Deroceras reticulatum* as measured by RNA-Seq (**A**) and quantitative real-time PCR (qRT-PCR) (**B**). The *y*-axis indicates the value of relative expression level (2^−ΔΔCt^) by qRT-PCR and log2 fold change of the nematode-infected slug (N-S) and the control slug (C-S).

**Table 1 insects-15-00311-t001:** Summary of the total number of bases, reads, and percentages of GC, Q20, and Q30 from the 6 samples.

Sample	Raw Reads	Raw_Bases	Valid Reads	Valid_Bases	Valid%	Q20%	Q30%	GC%
Control slug1	41,649,952	6.29 G	38,996,996	5.39G	93.63	97.81	93.85	37.50
Control slug2	45,331,020	6.84 G	42,417,080	5.87G	93.57	98.04	94.38	40.81
Control slug3	52,415,534	7.91 G	49,437,474	6.86G	94.32	98.05	94.33	40.31
Nematode-infected slug1	47,876,956	7.23 G	45,098,082	6.27G	94.20	97.86	94.17	39.40
Nematode-infected slug2	47,830,174	7.22 G	45,316,882	6.33G	94.75	98.05	94.15	37.55
Nematode-infected slug3	44,114,910	6.66 G	41,826,018	5.82G	94.81	98.08	94.28	37.89

**Table 2 insects-15-00311-t002:** Unigene annotation against NCBI databases.

Databases (DB)	Number of Unigenes	Ratio (%)
All	64,679	100.00
GO	7786	12.04
KEGG	7767	12.01
Pfam	7787	12.04
swissprot	7049	10.90
eggNOG	9618	14.87
NR	13,285	20.54
All	64,679	100.00

## Data Availability

The original contributions presented in the study are included in the article/Appendix A, further inquiries can be directed to the corresponding author.

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
