# Peer review of "Immune-Related Gene Profiles and Differential Expression in the Grey Garden Slug *Deroceras reticulatum* Infected with the Parasitic Nematode *Phasmarhabditis hermaphrodita"

_insects, 2024, doi:10.3390/insects15050311_

Round 1

Reviewer 1 Report

Comments and Suggestions for Authors

Review Report: insects-2961087

The study by Hafeez et al. explores the differential gene expression in the common terrestrial slug, Deroceras reticulatum, following nematode infection. This research is significant given the economic impact of slugs as pests in agriculture and the potential for utilizing pathogenic nematodes for pest control. The investigation aims to identify slug immune-related genes activated in response to nematode infection, shedding light on the mechanisms underlying slug immunity and providing insights for pest control strategies.

The researchers utilized RNA sequencing (RNA-seq) to analyze the gene expression profiles of nematode-infected slugs (N-S) compared to uninfected control slugs (C-S). The study identified a total of 39,380 regulated unigenes, with 3,084 up-regulated and 6,761 down-regulated in the N-S, indicating significant changes in gene expression upon nematode infection. Functional enrichment analysis, including gene ontology (GO), eukaryotic Ortholog Groups of proteins (KOG), and Kyoto Encyclopedia of Genes and Genomes Pathway (KEGG) databases, was performed to elucidate the biological functions of differentially expressed genes (DEGs). The study focused on approximately 228 up-regulated genes associated with immunity or immune-related pathways, including Toll, Imd, JNK, scavenger receptors (SCRs), C-type lectins (CTLs), immunoglobulin-like domains, and JAK/STAT63 signaling pathways. Furthermore, the expression levels of 18 selected genes were validated using quantitative real-time polymerase chain reaction (qRT-PCR), corroborating the RNA-seq findings.

The findings of this study provide valuable insights into the immune response of slugs during nematode infection. By identifying specific genes and pathways activated in response to nematode invasion, the research enhances our understanding of slug immunity mechanisms. Notably, the up-regulation of genes associated with Toll, Imd, JNK, and other immune-related pathways suggests a multifaceted immune response to nematode infection in slugs. This comprehensive analysis of DEGs contributes to the existing knowledge base regarding slug immunity and host-pathogen interactions.

The study's implications extend beyond basic research, offering practical applications for pest control in agriculture. By elucidating the immune response of slugs to nematode infection, the research provides a foundation for developing novel pest management strategies. The identification of immune-related genes and pathways activated in response to nematode invasion informs the selection and optimization of pathogenic nematodes for biological control of slug populations. Moreover, the validation of gene expression levels through qRT-PCR enhances the reliability and reproducibility of the findings, strengthening the study's impact and utility.

In conclusion, the study by Hafeez et al. represents a significant advancement in our understanding of slug immunity and its response to nematode infection. Through comprehensive RNA-seq analysis and functional enrichment studies, the research identifies key genes and pathways involved in the immune response of slugs. These findings not only contribute to the body of knowledge in immunology and host-pathogen interactions but also have practical implications for pest management in agriculture. Moving forward, further research may explore the potential applications of these insights in developing effective and sustainable strategies for pest slug control using pathogenic nematodes.

Comments to authors:

While the Materials and Methods section provides an overview of the experimental procedures, it lacks sufficient detail and clarity in several areas, potentially compromising the reproducibility and robustness of the study's results. Addressing these shortcomings through additional information and clarification would enhance the transparency and reliability of the research findings.

Lack of Detailed Information on Slug Collection and Handling: While the section mentions the collection of Deroceras reticulatum from grass seed production fields in the Willamette Valley, Oregon, USA, it lacks crucial details such as the specific locations, times of collection, and any relevant environmental factors. Additionally, details regarding slug handling during transportation from the collection sites to the laboratory are not provided, which could potentially impact the slugs' physiological condition upon arrival at the laboratory.

Inadequate Description of Infection Trial Protocol: The infection trial protocol is briefly mentioned, indicating the use of a single rate of P. hermaphrodita nematodes for infecting adult slugs. However, essential details such as the method of nematode application, duration of exposure, and criteria for determining successful infection are not sufficiently described. Moreover, while the mortality rate of slugs in the infection trial is mentioned, details regarding the monitoring of slug health and any observed symptoms of nematode infection are lacking.

Limited Information on Data Analysis Methods: While the section briefly describes the de novo transcriptome assembly and functional annotation processes, it lacks sufficient detail regarding specific software tools, parameter settings, and quality control measures employed during data analysis. Additionally, the criteria used for identifying immunity-related genes and pathways from the transcriptome data are not clearly outlined, making it challenging to assess the rigor and reproducibility of the gene annotation process.

Insufficient Explanation of RNA Isolation and Sequencing Procedures: while the sequencing platform (Illumina NovaSeq6000) is mentioned, essential details such as read lengths, sequencing depth, and quality control measures are not provided, which are crucial for assessing the reliability and robustness of the sequencing data.

Incomplete Statistical Analysis Description: Although the section mentions the use of student’s t-test for comparing mRNA expression levels between control and treatment groups, it lacks information on the specific statistical tests used to analyze other aspects of the data, such as differential gene expression analysis and validation of DEGs by qRT-PCR. Moreover, details regarding the handling of multiple testing corrections and any assumptions underlying the statistical analyses are not provided, which are essential for ensuring the validity of the study's findings.

After evaluating the manuscript, I recommend accepting it with minor revisions.

Author Response

Response to Reviewer 1 Comments

We appreciate your time with valuable comments! We responded and addressed all the comments (with bold letters), and revised the manuscript accordingly using track changes in the text.

Comments to authors:

 While the Materials and Methods section provides an overview of the experimental procedures, it lacks sufficient detail and clarity in several areas, potentially compromising the reproducibility and robustness of the study's results. Addressing these shortcomings through additional information and clarification would enhance the transparency and reliability of the research findings.

Lack of Detailed Information on Slug Collection and Handling: While the section mentions the collection of Deroceras reticulatum from grass seed production fields in the Willamette Valley, Oregon, USA, it lacks crucial details such as the specific locations, times of collection, and any relevant environmental factors. Additionally, details regarding slug handling during transportation from the collection sites to the laboratory are not provided, which could potentially impact the slugs' physiological condition upon arrival at the laboratory. Added more details with relevant refs. “Field-collected slugs were identified and confirmed as D. reticulatum using Vlach (2016) and Mc Donnell et al. (2009). Slugs were not being cultured in the laboratory. We have used field-collected slugs and nematode-infected slugs according to previously published methods and guidelines (Denver et al., 2024; Mc Donnell et al., 2022; 2020)”.

Inadequate Description of Infection Trial Protocol: The infection trial protocol is briefly mentioned, indicating the use of a single rate of P. hermaphrodita nematodes for infecting adult slugs. However, essential details such as the method of nematode application, duration of exposure, and criteria for determining successful infection are not sufficiently described. Moreover, while the mortality rate of slugs in the infection trial is mentioned, details regarding the monitoring of slug health and any observed symptoms of nematode infection are lacking. As the above, we described more details with relevant references cited.

Limited Information on Data Analysis Methods: While the section briefly describes the de novo transcriptome assembly and functional annotation processes, it lacks sufficient detail regarding specific software tools, parameter settings, and quality control measures employed during data analysis. Additionally, the criteria used for identifying immunity-related genes and pathways from the transcriptome data are not clearly outlined, making it challenging to assess the rigor and reproducibility of the gene annotation process. Added with ‘ We manually checked all putative hits, looking for the presence of the candidate genes and characteristic domains. The candidate transcripts were further searched against the NCBI database. Based on the expression values as FPKM, DEGs were identified between the two samples (N-S and C-S) after transformation, normalization and fold change (fc) comparisons’. to Section 2.4 and 2.7, respectively.  

Insufficient Explanation of RNA Isolation and Sequencing Procedures: while the sequencing platform (Illumina NovaSeq6000) is mentioned, essential details such as read lengths, sequencing depth, and quality control measures are not provided, which are crucial for assessing the reliability and robustness of the sequencing data. The six RNA samples (N-S and C-S per 3 replicates) were then sent to Psomagen (Rockville, MD, USA) for the RNA quality analysis using an Agilent 2100 Bioanalyzer (Agilent Technologies, Santa Clara, CA, USA). To maximize RNA quality, only samples with an RNA integrity number (RIN) value of 7 or greater were used for the next step. For cluster generation, the library is loaded into a flow cell where fragments are captured on a lawn of surface-bound oligos complementary to the library adapters. Each fragment is then amplified into distinct, clonal clusters through bridge amplification. When cluster generation is complete, the templates are ready for sequencing. The raw sequence reads generated by Illumina sequencing were checked by FastQC for quality control using through an integrated primary analysis software called RTA (Real Time Analysis).’

Incomplete Statistical Analysis Description: Although the section mentions the use of student’s t-test for comparing mRNA expression levels between control and treatment groups, it lacks information on the specific statistical tests used to analyze other aspects of the data, such as differential gene expression analysis and validation of DEGs by qRT-PCR. Moreover, details regarding the handling of multiple testing corrections and any assumptions underlying the statistical analyses are not provided, which are essential for ensuring the validity of the study's findings. Thanks for the comments. We used a t-test to analyze our data. In the paper, we focused on presenting the expression trend/pattern of immunity genes in the Discussion section, rather than just the statistical values.

After evaluating the manuscript, I recommend accepting it with minor revisions.

Reviewer 2 Report

Comments and Suggestions for Authors

Manuscript ID: insects-2961087- "Immune-Related Gene Profile and Differentially Expressions in the Grey Garden Slug Deroceras reticulatum Infected with the Parasitic Nematode Phasmarhabditis hermaphrodita" by Hafeez et al. investigated and analyzed the differential gene expressions profile of nematode-infected slugs, and compared them to the uninfected slugs, and identified the genes associated with immunity. The manuscript also determined biological functions of deferentially expressed genes (DEGs), gene ontology (GO) and functional enrichment analysis using eukaryotic Ortholog Groups of proteins (KOG) and Kyoto Encyclopedia of Genes and Genomes Pathway (KEGG) databases. Addressing the suggested improvements and major revisions would enhance the manuscript's impact and readability.

Whether field collected garden slugs have been identified/determined and validated for correct species selection by any mean? As they are not being culture in the laboratory.

Figure 1: uninfected and infected slugs “arrows” should be segregated; according to flow, it seems like pooling of samples.

Methods of the paper need more description like nematode worm infection and explaining the default setup of used software would be helpful to other user to replicate the same exercise elsewhere.

Section 2.5: line # 159-161; it would be good if you here disclose if any genome assembly of the organism (Deroceras reticulatum) are available or not?

Figure 5 and 6: y-axis is it N-S or S-N? There is a discrepancy in text and figures/figure legend.

These are some nice articles where immune pathway genes from STAT signaling, and dual oxidase and heme peroxidase gene expressions have been reported (26407822; 26943999 and 28352267) Discussion of these would help the to strengthen the RT-qPCR results and better understand how the current study contributes to the field of innate immunity.

What kind of statistical analysis have been performed; there is no mention in the legends and/or in figures.

Institutional review board and consent section should have some comment in their respective place. The manuscript should include a section that highlights the ethical considerations taken into account, particularly with respect to data accession/collection date and availability of public domain data, as the assembly used to get change periodically.

Section 2.7: Write clearly Table S1 and Table S2, wherever used in the manuscript. Similarly, in section 3.5, number in parentheses are the ratio or number of genes?

The manuscript has done comprehensive/exhaustive review of the existing literature on different dynamics. Citing in methods papers like PMID: 27664587; and PMID: 34578158 would be useful to understand the bio-informatics analysis.

The manuscript's language and writing style are generally clear and concise. However, some sentences appear to be lengthy (Line # 254-258) and could be restructured for better understanding and readability.

Some spellings of the taxonomical species are not correct. I would suggest double check that all over the manuscript. Overall, I would recommend justify all these small mistakes at your end; above are some examples.

All the very best.

Comments on the Quality of English Language

Authors should take care of rest likewise and proof-read (English and Grammar) at your end.

Author Response

Response to Reviewer 2 Comments

We appreciate your time with valuable comments! We responded and addressed to all the comments (with bold letters), and revised the manuscript accordingly using track changes in the text.

Comments and Suggestions for Authors

Manuscript ID: insects-2961087- "Immune-Related Gene Profile and Differentially Expressions in the Grey Garden Slug Deroceras reticulatum Infected with the Parasitic Nematode Phasmarhabditis hermaphrodita" by Hafeez et al. investigated and analyzed the differential gene expressions profile of nematode-infected slugs, and compared them to the uninfected slugs, and identified the genes associated with immunity. The manuscript also determined biological functions of deferentially expressed genes (DEGs), gene ontology (GO) and functional enrichment analysis using eukaryotic Ortholog Groups of proteins (KOG) and Kyoto Encyclopedia of Genes and Genomes Pathway (KEGG) databases. Addressing the suggested improvements and major revisions would enhance the manuscript's impact and readability.

Whether field collected garden slugs have been identified/determined and validated for correct species selection by any mean? As they are not being culture in the laboratory. Added more details with relevant refs. Field collected slugs were identified and confirmed as D. reticulatum using Vlach (2016) and Mc Donnell et al. (2009). Slugs were not being cultured in the laboratory. We have used field-collected slugs and nematode-infected slugs according to previously published methods and guidelines (Denver et al., 2024; Mc Donnell et al., 2022; 2020).

Figure 1: uninfected and infected slugs “arrows” should be segregated; according to flow, it seems like pooling of samples. Thanks for the comment. We revised Figure 1 and made it clear.

Methods of the paper need more description like nematode worm infection and explaining the default setup of used software would be helpful to other user to replicate the same exercise elsewhere. Added more description on the collection of the nematode-infected slugs 

Section 2.5: line # 159-161; it would be good if you here disclose if any genome assembly of the organism (Deroceras reticulatum) are available or not? Thank you for the comment, the raw data of the transcriptome are not yet ready to publish because we are working to find other target genes. S1 excel file provided in this study will facilitate to find all immune-related genes in the slug

Figure 5 and 6: y-axis is it N-S or S-N? There is a discrepancy in text and figures/figure legend. Corrected through the text and figures.

These are some nice articles where immune pathway genes from STAT signaling, and dual oxidase and heme peroxidase gene expressions have been reported (26407822; 26943999 and 28352267) Discussion of these would help the to strengthen the RT-qPCR results and better understand how the current study contributes to the field of innate immunity. What kind of statistical analysis have been performed; there is no mention in the legends and/or in figures. Thanks for the refs introduced, we added and discussed with those refs. We used a t-test to analyze our data. In the paper, we focused on presenting the expression trend/pattern of immunity genes in the Discussion section, rather than just the statistical values.

Institutional review board and consent section should have some comment in their respective place. The manuscript should include a section that highlights the ethical considerations taken into account, particularly with respect to data accession/collection date and availability of public domain data, as the assembly used to get change periodically. The original data and contributions of the study are included in the article/supplementary material, and further inquiries can be directed to the corresponding author upon request.

Section 2.7: Write clearly Table S1 and Table S2, wherever used in the manuscript. Similarly, in section 3.5, number in parentheses are the ratio or number of genes? Described more and added the number of genes in parentheses

The manuscript has done comprehensive/exhaustive review of the existing literature on different dynamics. Citing in methods papers like PMID: 27664587; and PMID: 34578158 would be useful to understand the bio-informatics analysis. Thank you for the comments, we cited the paper.

The manuscript's language and writing style are generally clear and concise. However, some sentences appear to be lengthy (Line # 254-258) and could be restructured for better understanding and readability. Made these sentences are concise for better readability.

Some spellings of the taxonomical species are not correct. I would suggest double check that all over the manuscript. Overall, I would recommend justify all these small mistakes at your end; above are some examples. Thank you for the comments, we checked again through the paper.

Round 2

Reviewer 2 Report

Comments and Suggestions for Authors

Manuscript ID: insects-2961087 peer-review-v2- "Immune-Related Genes Profile and Differentially Expressions in the Grey Garden Slug Deroceras reticulatum Infected with the Parasitic Nematode Phasmarhabditis hermaphrodita" by Hafeez et al. has substantially proof-read and amended the manuscript. Although, naturally infected and field collected slug species papers you have published before but here you are doing whole-exome sequencing looks good to me.

In general, the organization and the structure of the article are satisfactory and in agreement with the journal instructions for authors. The work shows a conscientious study in which a very exhaustive discussion of the literature available has been carried out. The introduction provides sufficient background, and the other sections clearly presented and analyzed exhaustively. I am expecting that author will take care of other formatting errors and English language errors at their end.

All the very Best.

Comments on the Quality of English Language

I am expecting that author will take care of other formatting errors and English language errors at their end.